# Mesenchymal Stem Cell Therapy in Diabetic Cardiomyopathy

**DOI:** 10.3390/cells11020240

**Published:** 2022-01-11

**Authors:** Jaqueline S. da Silva, Renata G. J. Gonçalves, Juliana F. Vasques, Bruna S. Rocha, Bianca Nascimento-Carlos, Tadeu L. Montagnoli, Rosália Mendez-Otero, Mauro P. L. de Sá, Gisele Zapata-Sudo

**Affiliations:** 1Programa de Pesquisa em Desenvolvimento de Fármacos, Instituto de Ciências Biomédicas, Universidade Federal do Rio de Janeiro, Av. Carlos Chagas Filho, 373, Rio de Janeiro 21941-902, RJ, Brazil; ssjck@hotmail.com (J.S.d.S.); brunadesouzarocha.98@gmail.com (B.S.R.); biascarlos@yahoo.com.br (B.N.-C.); tmontagnoli@gmail.com (T.L.M.); 2Instituto do Coração Edson Saad, Faculdade de Medicina, Universidade Federal do Rio de Janeiro, Street Prof. Rodolpho Paulo Rocco, 255, Rio de Janeiro 21941-617, RJ, Brazil; paesleme@hucff.ufrj.br; 3Instituto de Biofísica Carlos Chagas Filho, Universidade Federal do Rio de Janeiro, Av. Carlos Chagas Filho, 373, Rio de Janeiro 21941-170, RJ, Brazil; renata.guedes@biof.ufrj.br (R.G.J.G.); rmotero@biof.ufrj.br (R.M.-O.); 4Instituto de Ciências Biomédicas, Universidade Federal do Rio de Janeiro, Av. Carlos Chagas Filho, 373, Rio de Janeiro 21941-170, RJ, Brazil; juliana.vasques@icb.ufrj.br; 5Instituto Nacional de Ciência e Tecnologia em Medicina Regenerativa, Av. Carlos Chagas Filho, 373, Rio de Janeiro 21941-902, RJ, Brazil

**Keywords:** diabetes mellitus, cardiac remodeling, diabetic cardiomyopathy, mesenchymal stromal cells, pro-inflammatory cytokines, oxidative stress, fibrosis

## Abstract

The incidence and prevalence of diabetes mellitus (DM) are increasing worldwide, and the resulting cardiac complications are the leading cause of death. Among these complications is diabetes-induced cardiomyopathy (DCM), which is the consequence of a pro-inflammatory condition, oxidative stress and fibrosis caused by hyperglycemia. Cardiac remodeling will lead to an imbalance in cell survival and death, which can promote cardiac dysfunction. Since the conventional treatment of DM generally does not address the prevention of cardiac remodeling, it is important to develop new alternatives for the treatment of cardiovascular complications induced by DM. Thus, therapy with mesenchymal stem cells has been shown to be a promising approach for the prevention of DCM because of their anti-apoptotic, anti-fibrotic and anti-inflammatory effects, which could improve cardiac function in patients with DM.

## 1. Introduction

The incidence and prevalence of diabetes mellitus (DM) worldwide are increasing rapidly, with 19.3% of the population aged 65–99 diagnosed with DM [1,2], and type 2 (DM2) accounts for 90–95% of cases [3]. According to the World Health Organization, DM was the ninth cause of global death in 2019, with cardiovascular complications being the main cause of the deaths [4]. Among the cardiac complications is diabetic cardiomyopathy (DCM), which consists of cardiac remodeling characterized by molecular, cellular and interstitial changes, interfering with the size, geometry and function of the heart. DCM includes cell death, changes in energy metabolism and the extracellular matrix, inflammation, oxidative stress, neurohormonal activation and modification in ion transport [5]. DM [6,7,8,9,10,11] can induce cardiac changes due to remodeling and culminate in cardiac dysfunction, which is independent of other cardiac comorbidities, such as coronary artery disease, hypertension and valvular heart disease [8].

Traditional DM treatment, which can interfere with insulin sensitization and secretion, in addition to causing undesirable side effects [12], fails to prevent and/or treat cardiac remodeling. In recent years, cell therapy is one approach that has been used mainly due to its paracrine action, producing immunomodulation and anti-inflammatory and anti-apoptotic activities.

Mesenchymal stem cells (MSCs) are multipotent cells found in almost all adult tissues with the abilities of self-renewal and differentiation, favoring homeostasis and repair, which make them promising tools for regenerative medicine [13]. MSCs show low immunogenicity, which can allow allogeneic cell transplantation, and are home to damaged tissues, where they can engraft and differentiate, promoting repair. However, MSCs have a poor engraftment rate, and cellular replacement is complex and limited. Instead, their therapeutic potential is attributed primarily to the paracrine action of their secretome, which consists of a rich and complex mixture of soluble molecules, such as cytokines, chemokines and growth factors, and extracellular vesicles loaded with proteins, peptides and genetic material (e.g., microRNAs) [14], which can support cell survival and tissue healing. Additionally, preconditioning with hypoxia, growth factors, cytokines or pharmacological agents and genetic modifications can modulate MSCs’ survival, proliferation, migration and senescence in order to preserve them and improve their secretory activity, amplifying their therapeutic efficacy [14].

Thus, the present review focuses on the mechanisms involved in DM-induced cardiac remodeling and, consequently, DCM and the potential of MSC therapy to treat DM complications.

## 2. Molecular Mechanisms Involved in Diabetes-Induced Cardiac Remodeling

In the early stages, DCM is characterized by cardiomyocyte apoptosis and myocardial fibrosis with increased deposition of accumulated extracellular matrix (ECM), which leads to rigidity in the diabetic heart, exerting a deleterious effect on diastolic function causing diastolic heart failure with preserved ejection fraction (HFpEF) [15]. Exceeding collagen deposition and the abnormal alignment of cardiomyocytes increase myocardial rigidity resulting from DM and affect systolic function, which may progress to heart failure with reduced ejection fraction (HFrEF) [16,17]. A high glucose level in DM is responsible for myocardial dysfunction because of the increased fatty acid metabolism, reduced sensitivity to Ca^2+^ by myofilaments, mitochondrial dysfunction, oxidative stress [11,15,18,19,20], high levels of advanced glycation end products (AGE) [21], increased inflammation, apoptosis and necrosis [11,22]. These altered properties lead to an imbalance between cell death and survival [6], which, together with increased collagen, fibrosis and hypertrophy [18], culminate in cardiac remodeling.

Hyperglycemia promotes the formation of reactive oxygen species (ROS) and reactive nitrogen species (RNS) [6,7]. ROS induce apoptosis [6] via cytochrome-c-mediated caspase-3 activation [7], which modulates the growth and survival of multipotent cardiac stem cells and progenitor cells (CSPCs) in the heart. Hyperglycemia-induced oxidative stress promotes the increase in mitochondrial ROS production, which triggers cell damage pathways involved in CSPC loss, which, in turn, increases cell death [23,24]. DM induces an increase in the apoptosis/necrosis of myocytes and CSPC, in addition to a reduction in CSPC proliferation, resulting in cardiac myopathy with impaired ventricular function. Oxidative stress is responsible for mitochondria and sarcoplasmic reticulum structural changes, favoring the increase in cell death in animal models of DM [7]. A high plasma superoxide dismutase (SOD) level is detected in DM, which is not accompanied by an increase in plasma glutathione peroxidase (GSH-Px), leading to an increase in oxidative stress and consequent uncontrolled lipid peroxidation [9]. Thus, ROS and malondialdehyde (MDA), a stable end product of lipid peroxidation, are increased in DM, indicating failure of the antioxidant system [25,26,27], which can result in damage to the cells (Figure 1). The reduction in ROS production and the improvement in antioxidant defense can be promoted by sirtuin 1 (SIRT1), which has nicotinamide adenine-dependent deacetylase activity [28,29]. Therefore, the regulation of oxidative stress, inflammation and cardiomyocyte survival by SIRT1 is impaired, whereas cardiac SIRT1 expression is markedly decreased in DM [27,30,31].

Increased expression of NADPH oxidase 2 (NOX2) and inducible nitric oxide synthase (iNOS) are observed in cardiomyocytes exposed to high levels of glucose and fat, with consequent increase in ROS/RNS formation and cytochrome-c release, leading to cell death (Figure 1). In DM, the levels of full-length type III fibronectin containing 5 (FNDC5) have been reported to be reduced, and they are involved in cardiomyocyte apoptosis [8,32]. The overexpression of FNDC5 reduces NOX2 and iNOS activities, improves glucose tolerance, reverses diastolic dysfunction and attenuates cardiac remodeling, demonstrating a significant role of reduced FNDC5 in cardiac dysfunction observed in DM [8].

High levels of ROS and inflammatory mediators due to hyperglycemia can promote the synthesis of highly reactive carbonyl intermediates from lipid peroxidation, such as glyoxal and methylglyoxal, which lead to the formation of AGE [33]. The activation of AGE/AGE receptor (RAGE) signaling promotes increased ECM synthesis and accumulation in DM [34]. This is due to the increased expression and secretion of various types of collagen and pro-fibrotic factors promoted by AGE, generating ECM crosslinking, which alters the cell–matrix interaction and cell adhesion, as well as modifying the regulation of oxidative stress and inflammation [35]. The AGE–RAGE interaction induces the formation of ROS and activates other signaling proteins, such as extracellular signal-regulated kinases (ERK) 1/2, which, in turn, increase the expression and phosphorylation of activated B cell nuclear factor kappa (NF-κB). Furthermore, AGE can alter SOD expression and impact the expression of proteins related to ECM remodeling [21]. Thus, the increase in ROS, the accumulation of catechol-modified proteins, the expression and secretion of type I and type III collagen and pro-fibrotic activation markers (such as STAT3) explain the occurrence of extracellular matrix remodeling [36]. Increased AGE has also been implicated in fibroblast activation and proliferation in DM [37]. Fibroblasts are closely related to cardiac remodeling, since there is an increase in the expression of α-smooth muscle actin (α-SMA), suggesting the transformation of fibroblasts into a myofibroblast phenotype [38].

P2X7 purinergic receptor (P2X7R) signaling has also been implicated in cardiac remodeling in DM through the modulation of fibrosis and hypertrophy [39]. P2X7R exerts important physiological and pathological effects on the cardiovascular system [40]; its stimulation regulates pro-inflammatory responses in endothelial cells [41,42]. The upregulation of P2X7R in the hearts of diabetic mice and in cardiomyocytes exposed to high glucose content appears to be responsible for the increased activation of protein kinase C β (PKC β) and ERK, collagen I and TGF-β, caspase-3 and Bax. The inhibition of P2X7R significantly reduces cardiac damage, including fibrosis, apoptosis and hypertrophy, in vivo and in vitro, possibly by reducing the inhibition of the PKC/ERK pathway [39].

Thus, it is clear that the oxidative stress generated by DM promotes cardiac remodeling acting through different signaling pathways in both cardiomyocytes and cardiac fibroblasts, which is a target for the prevention or treatment of DCM.

## 3. Myocardial Inflammation and Diabetes-Induced Cardiac Remodeling

Metabolic disorders, such as DM, induce a systemic inflammatory state, which is reflected in myocardial structure and function [43], progressively leading to a clinically evident cardiomyopathy phenotype, such as HFpEF [44]. Thus, chronic inflammation leads to metabolic reprogramming of the heart and contributes to adverse remodeling and functional impairment [45].

Initially, this proinflammatory state is stimulated by high levels of blood glucose and free fatty acids, acting on different cell types, including resident cardiomyocytes, fibroblasts and macrophages, and, when stimulated, it converges to activate NF-κB and increase the cytokine expression of TNF-α and IL-6 [46]. Macrophage infiltration in diabetic hearts is associated with a high level of vascular adhesion molecules, such as intercellular adhesion molecule (ICAM)-1 and vascular cell adhesion molecule (VCAM)-1 [43,47], which are related to increased cardiac cellular infiltrate, as well as to a reduction in the concentration of cardiac IL-10 [48]. A high glucose concentration directly activates several pro-inflammatory pathways in cardiac fibroblasts, macrophages and cardiomyocytes due to the induction of the sustained transcription of high mobility protein box 1 (HMGB1) followed by increased NF-κB binding activity with an increased and sustained expression of TNF-α, JNK, ERK1/2 and IL-6 [49]. Furthermore, in chronic hyperglycemia, AGE can accumulate in cardiac ECM and increase the expression of pro-inflammatory mediators, including TNF-α, IL-6, ICAM-1 and CC motif chemokine ligand 2 (CCL2) [43]. AGE also promotes myocardial inflammation by the direct activation of macrophages, promoting the increased expression of TNF-α, iNOS and IL-6 through the RAGE/NF-κB pathway [50,51].

Furthermore, in spontaneously insulin-resistant rats, an increase in the levels of several inflammatory cytokines was observed, such as IL-2, IL-6, IL-1β, IL-4, IL-5, IL-12p70, stimulating factor granulocyte and macrophage colonies (GM-CSF), interferon gamma (IFNγ) and TNF-α. These cytokines can increase the QT interval in diabetic animals, indicating cardiac electrical remodeling. Kv1.3 potassium channels were involved in this process, since treatment with a channel blocker reverses changes in the cardiomyocytes of diabetic rats [52]. Similarly, it was demonstrated in diabetic animals that an electrical remodeling impairs the repolarization of cardiac myocytes due to the increase in TNF-α and IL-1β, confirmed by the prevention of the prolongation of the cardiac action potential by the blockade of TNF-α and IL-1β receptors [53].

Transforming growth factor beta (TGF-β) is the main factor involved in cardiac fibrosis, since its expression is increased in cardiomyocyte cells in hyperglycemic conditions, and the increased expression of TGF-β is related to a significant increase in collagen synthesis and increased ECM deposition [54]. A possible mechanism proposed for the increase in TGF-β in DM is the activation of chemokine receptor type 4 (CXCR4), since the inhibition of this receptor reduces TGF-β levels [11]. TGF-β/Smad signaling could act via endothelial to mesenchymal cell transition (EndMT), playing an important role in the pathogenesis of cardiac fibrosis in DM [55,56]. EndMT is an important pathway for changing cell phenotype, promoting the proliferation of cardiac fibroblasts and aggravating cardiac fibrosis in the presence of hyperglycemia [57,58]. In the heart of diabetic rats, upregulation of mesenchymal markers (α-SMA, vimentin and fibroblast-specific protein-1) and downregulation of endothelial markers (CD31 and vascular endothelial cadherin) in peripheral vessels, as well as the reduced phosphorylation of AMPKα, are detected in cardiac tissue. The transcription level of EndMT markers (snail 1, snail 2, twist 1 and twist 2) was also elevated in diabetic rat hearts. Importantly, inhibition of the TGF-β/Smad pathway can be achieved by dapagliflozin and metformin through AMPKα activation, with the suppression of cardiac fibroblast activity [59].

## 4. Epigenetic Involvement in Diabetic Cardiomyopathy

The influence of metabolic status on gene expression is well known, and research has recently provided mechanistic insights into the association between epigenetic regulation and cardiac pathological remodeling in DCM [60,61]. Epigenetic mechanisms involve nucleotide and histone modifications, as well as other transcriptional and translational regulation by non-coding RNAs. These modifications exert long lasting effects, which subside a “hyperglycemic memory” observed in cells from DM patients and animal models [60,62] and may increase the transgenerational risk to cardiometabolic diseases [62,63].

Direct DNA methylation modulates the accessibility of gene promoter sequences to transcription factors, often resulting in gene repression. Cardiomyocytes from diabetic rats display alterations in DNA methylation patterns of cyclin D1 and p21^WAF1/CIP1^ genes when submitted to oxidative stress, which result in increased cell death [64]. Recently, the suppression of androgen receptor transcription was associated with high DNA methyltransferase-1 activity in cardiac fibroblasts in a DM rodent model, which resulted in elevated autophagy and collagen synthesis [65].

Histone methylation, acetylation and phosphorylation are also important mechanisms implicated in gene transcription regulation. Increased activity of histone acetyltransferase (HAT) p300 is observed in rat models of DCM and correlates to the severity of cardiac dysfunction, myocardial hypertrophy and fibrosis [66,67]. The influence of histone acetylation on DCM seems to involve the recruitment of transcriptional regulators, since the treatment with an inhibitor of bromodomain-containing protein-4 (BRD4), apabetalone, decreased the risk of major adverse cardiovascular events and the frequency of hospitalizations after recent acute coronary syndrome in DM2 patients [68,69]. In addition to this, BRD4 is overexpressed in DM rodent models, which leads to increased gene expression of hypertrophy and fibrosis markers along with mitochondrial and cardiac dysfunction [70,71].

The pharmacological activation of sirtuins, promoting histone deacetylation, is also reported to attenuate cardiac changes observed in DCM by preventing ROS production and mitophagy, collagen deposition and apoptosis in DM models [62,72,73,74,75,76,77]. However, the inhibition of other histone deacetylase (HDAC) isoforms is reported to reduce the expression of pro-hypertrophic, pro-inflammatory and pro-fibrotic genes in different models of DCM [74,78,79,80,81,82,83]. The involvement of histone methylation in controlling gene expression in DCM by modulating cytokine production, gap junction integrity, mitochondrial dysfunction, metabolic imbalance, cardiac cell death and tissue inflammation is certain [84,85,86]. Along with transcriptional regulation, the role of long non-coding RNAs and micro-RNAs in the modulation of cardiac protein expression in DCM is a topic of great interest [87,88,89,90]. Epigenetic mechanisms act in different regulatory pathways in order to control gene expression. Regulation of JunD and p66^Shc^, consequent to changes in promoter and histone methylation patterns, plays an important role in experimental and human DCM [91,92,93].

## 5. Preclinical and Clinical Approaches to MSC in Diabetes-Induced Cardiac Complications

MSCs were originally obtained from bone marrow [94], which is still the most used source of MSCs for cell-based therapy in preclinical and clinical trials in a wide range of diseases [95]. However, they can also be isolated from multiple sources, including fat, skin, muscle and the heart. In recent years, MSCs derived from neonatal tissues (the placenta and umbilical cord) have been widely explored in cell therapy research, as they can be easily and non-invasively obtained with no pain or risk [96]. MSCs are identified by a set of standard criteria, which ascertain their equivalence: (1) plastic adherence under standard culture conditions; (2) expression of CD105, CD73 and CD90 surface markers and lack of expression of hematopoietic progenitor markers CD45, CD34, CD14/CD11b, CD79α/CD19 and HLA-DR; and (3) differentiation into osteoblasts, adipocytes and chondroblasts [97].

MSCs can differentiate into insulin-producing cells (IPCs) both in vitro and in vivo, which underlies their potential for treating DM [98]. The transplantation of IPCs derived from different types of MSCs in vitro is effective in regulating blood glucose levels in experimental diabetic animals [99,100,101]. However, although MSCs migrate to the pancreas, the differentiation into IPCs is minimal, and it is not the explanation for the possible regenerative effects induced by cell therapy. On injured pancreatic islets, beneficial effects are consequent to the immunomodulatory, anti-inflammatory and anti-apoptotic actions of the molecules secreted by MSCs, which protect endogenous β-cells, stimulate their regeneration and proliferation and ameliorate insulin resistance, reversing hyperglycemia, the main determinant of DCM [102]. Regarding cardiac repair, MSCs have protective and regenerative effects in several types of cardiovascular diseases [103]. They can differentiate into cardiomyocytes, endothelial cells and vascular smooth muscle cells [104], but, again, rather than cellular replacement, tissue repair seems mainly related to MSCs’ secretome, through which they suppress inflammatory responses and induce neovascularization, resulting in cardiac tissue preservation [105]. In addition, MSCs also have an anti-fibrotic effect, decreasing cardiac fibroblast activity and collagen deposition, and they promote endogenous regeneration by stimulating cardiac resident stem cells and cardiac progenitors. Therefore, the anti-diabetic properties of MSCs combined with their cardioprotective features make their use promising for the treatment of DCM. The major effects of MSCs therapy for DCM are illustrated in Figure 2.

### 5.1. Adult Tissue-Derived MSCs

Most preclinical studies regarding MSCs as therapeutic agents for DCM use MSCs isolated from bone marrow (BM-MSCs). In one of the first reports, a single intravenous injection of 5 × 10^6^ BM-MSCs attenuated cardiac fibrotic remodeling and induced angiogenesis in streptozotocin (STZ)-induced diabetic rats [106]. BM-MSCs transplantation induced a significant improvement in several cardiac parameters in relation to the control group, such as left ventricular posterior wall thickness, fractional shortening (FS) and relative wall thickness. BM-MSCs therapy induced a reduction in cardiac fibrosis, an effect possibly induced by the modulation of metalloproteinase expression and activity. The increased arteriolar density in myocardium from BM-MSCs-treated groups indicated an increase in angiogenesis. The presence of transplanted cells in the myocardium 4 weeks after the injections, as well as co-localization with cardiac markers (such as troponin T), suggested that part of BM-MSCs’ beneficial effects might be due to their differentiation in cardiomyocytes [106]. In a model of obesity-induced DCM in mice, cardiac dysfunction is only detectable after a pharmacological stress (stimulation with dobutamine, a sympathomimetic drug); however, intravenous administration of BM-MSCs had no effect on cardiac function [107]. Other mechanisms of BM-MSCs’ positive effects in DCM have been proposed and investigated in recent years. Serial intravenous injections of 5 × 10^5^ BM-MSCs, once a week for four weeks, in addition to improving cardiac function as described above, have been shown to reduce the expression of the apoptotic protein caspase-3 in DCM rats’ myocardium, while levels of 14-3-3 and phospho-Ask1, proteins with anti-apoptotic activity, were increased by therapy [108]. Most of the effects induced by MSCs are due to their paracrine action, and MSCs-derived extracellular vesicles, which include microvesicles and exosomes, are in focus as a cell-free therapeutic alternative. BM-MSCs exosomes injected intravenously once a week for 12 consecutive weeks in diabetic rats significantly reduced left ventricular collagen levels, an important indicator of cardiac fibrosis. Transforming growth factor-beta (TGF-β)/Smad2 is one of the major signaling pathways related to cardiac fibrosis, and mRNA levels were upregulated in the myocardium of diabetic rats, and this effect was partially reverted by exosome therapy [109].

When considering a possible translation to clinical practice, it is important to evaluate the interactions between MSCs transplantation and the standard diabetes therapeutic strategies. In a recent work, Ammar and colleagues investigated the effects of BM-MSCs therapy in STZ rats that were also treated with metformin [110], currently the most used antidiabetic drug. Metformin reduced the cardioprotective effects and glucose control induced by intravenous BM-MSCs therapy. According to others, BM-MSCs therapy significantly reduced collagen deposition in the myocardium of STZ rats to levels similar to those of non-diabetic animals, but this effect was attenuated when also treated orally with metformin. BM-MSCs induced an increase in myocardium microvessel number, indicating an angiogenic effect, which was diminished but not abolished by metformin supplementation. The reduction in the beneficial effects of BM-MSCs therapy by metformin are probably related to the reduced homing and engraftment capacity of BM-MSCs in the myocardium caused by the drug [110]. Another important aspect considering clinical translation is the characterization and homogeneity of cell products. Recently, the effects of syndecan-2^+^ BM-MSCs have been analyzed in early pre-fibrotic DCM in db/db transgenic mice. Syndecan-2 is a heparan sulfate proteoglycan and is considered a potential functional marker for the isolation of an MSC subpopulation. Although syndecan-2^+^ BM-MSCs, syndecan-2^−^ BM-MSCs and wild-type BM-MSCs induced a reduction in cardiomyocyte stiffness and an increase in arteriole density, the effects were slightly less pronounced in the syndecan-2^+^ BM-MSC group [111].

In recent years, adipose tissue has been the most frequently used source of MSCs for clinical application because it provides not only a greater amount of cells compared to bone marrow but also allows autologous transplantation [112]. Recently, adipose tissue-derived MSCs (AT-MSCs) have been tested for the treatment of STZ-induced DCM in rats. Intravenous injection of allogeneic AT-MSCs reduced myocardial inflammation by promoting M2 macrophage polarization and reversing the elevation of the pro-inflammatory cytokines IL-6 and TNF-α in heart tissue. This strategy also ameliorated fibrosis by suppressing the proliferation of cardiac fibroblasts, reducing TGF-β expression and collagen deposition, preventing ventricular hypertrophy and diastolic dysfunction [113]. Pretreatment with cyclooxygenase-2 inhibitor or the genetic deletion of prostaglandin E2 synthase abolished the cardioprotective effects, indicating the involvement of prostaglandin secretion [114]. Intravenous autologous AT-MSCs transplantation reduces fibrosis and cell death in cardiac tissue by increasing the expression of survival signaling molecules and reducing hypertrophic markers levels, which results in improved cardiac function [115]. Recently, it was demonstrated that AT-MSCs might prevent DCM by downregulating the adhesion of macrophages in heart tissue and by inhibiting autophagy-related apoptosis of myocardial cells [116].

### 5.2. Preconditioning and Genetic Modification of Adult Tissue-Derived MSCs

Although in preclinical studies, BM-MSCs and AT-MSCs are usually isolated from young and healthy donors, in clinical practice, cell therapy is generally autologous, which means that cells are harvested from the patient a few weeks prior to transplantation, reducing the immunological rejection risk and related complications. However, DM can markedly alter the biological properties of endogenous MSCs [117,118]. Diabetic BM-MSCs have normal morphology but decreased proliferation rates and secrete less vascular endothelial growth factor (VEGF) and insulin-like growth factor-1 (IGF-1), cytokines related to angiogenesis and apoptosis inhibition, when compared to BM-MSCs from healthy donors. Moreover, diabetic BM-MSCs are more susceptible to apoptosis induced by hypoxia and serum deprivation and have impaired in vitro myogenic differentiation capacity [119]. However, some alterations seen in diabetic MSCs can be reverted using myogenic medium preconditioning. The preconditioning of BM-MSCs can improve their proliferative rate and the production of cardioprotective and angiogenic factors. This condition can also provide an increase in cardiac homing and a reduction in cardiac dysfunction and remodeling [117].

As previously mentioned, interesting approaches to increase MSCs’ therapeutic effects are preconditioning, changing culture conditions and genetically modifying cells in vitro prior to transplantation. BM-MSCs cultivated in anoxic conditions for 3h (AP-MSCs) prior to direct transplantation in the myocardium induced more expressive positive effects in diabetic rats than control BM-MSCs. AP-MSCs induced a greater improvement in heart hypertrophy and FS in relation to BM-MSCs kept in standard conditions. AP-MSCs also increased capillary density in the myocardium [120]. Improvement in stem cell survival and therapeutic potential was detected after intravenous administration of 1 × 10^6^ BM-MSCs combined with resveratrol (RSV-BM-MSCs) in diabetic rats. RSV has been associated with cardioprotective effects in DCM preclinical models [121], and RSV increased the proliferation rate and antioxidant properties of BM-MSCs. In vivo, long-term treatment with systemic RSV improved lipid profile, thereby reducing triglycerides levels. This was also observed in animals that received RSV-BM-MSCs but not in those submitted to cell therapy alone. Interestingly, animals treated with RSV-BM-MSCs presented better cardioprotective effects because echocardiography indicated that the values of ejection fraction (EF) and FS were similar to those of non-diabetic rats. Moreover, it reduced cardiac fibrosis and cardiomyocytes hypertrophy [122].

The previously described beneficial effects obtained with AT-MSC therapy were even greater when preconditioned with resveratrol, which increased cell viability, migration and secretory capacity by upregulating the expression of CXCR4 and proteins related to the antioxidant Sirt 1/Akt axis. They have also shown that the beneficial effects of cell therapy with AT-MSCs were potentiated by the oral administration of green tea epigallocatechin-3-gallate, which has great antioxidant potential [123].

Recently, BM-MSCs modified to express adiponectin (APN-BM-MSCs) were evaluated in heart fibrosis attenuation in rats submitted to a high-fat diet associated with STZ injection. APN is usually secreted by adipocytes and has been related to a reduction in insulin-resistance and anti-atherosclerotic effects. Intravenous injection of 4 × 10^6^ APN-BM-MSCs attenuated cardiac hypertrophy seen in diabetic rats, an effect not obtained with non-modified BM-MSC therapy. Echocardiograms demonstrated an improvement in left ventricular ejection fraction (LVEF) in animals treated with non-modified BM-MSCs, but other cardiac function parameters, such as fractional shortening (FS) and left ventricular internal diameter end systole (LVID), were only improved in rats that received APN-BM-MSCs. Regarding myocardium tissue morphology, APN-BM-MSC therapy reduced fibrotic area and collagen deposition. TGF-β)/Smad2/3, which was upregulated in diabetic hearts, was prevented in animals submitted to APN-BM-MSC therapy [124].

### 5.3. Perinatal Tissue-Derived MSCs

MSCs derived from birth-associated tissues represent a good alternative for allogeneic transplantation. In addition to the previously mentioned advantages in obtaining these types of cells, they display improved features over adult MSCs in terms of proliferation, life span and epigenetic modifications [125]. Intravenous injection of placental-derived MSC-like cells, termed PLX, improved diastolic function at an early stage of DCM in STZ-induced diabetic mice by improving left ventricle vascularization and reducing cardiomyocyte stiffness before the onset of myocardial inflammation and fibrosis [126]. Table 1 summarizes the main findings of MSC therapy in relation to cardiac structure and function in preclinical models of DM.

### 5.4. Clinical Studies

DCM is usually an asymptomatic disease in the early stages and is often accompanied by other complications induced by DM, making an accurate diagnosis and even the selection of patients for clinical trials difficult. Indeed, many of the DCM-related studies registered on clinicaltrials.gov (accessed on 21 December 2021) look for early, specific and effective diagnostic methods (NCT04593173, NCT04534894, NCT01295385, NCT01220349 and NCT04303364). To date, there are no clinical trials testing the efficacy of MSC-based therapy in the treatment of DCM specifically. However, there are several studies that include patients with ischemic and non-ischemic cardiomyopathies with DM as a comorbidity, most of them using BM-MSCs transplanted by transendocardial injection, although intravenous and intracoronary infusion have also been tested [127,128]. The majority have focused on the impact of cell therapy on ischemic cardiomyopathy, but a comparative analysis indicated that cardiac function improved preferentially in non-ischemic cardiomyopathy [129], with higher clinically significant efficacy for allogeneic than for autologous cells [130,131].

In general, cell therapy is safe and feasible for treating cardiomyopathy, with no serious adverse events and recovered parameters of systolic and diastolic functions and functional capacity, improving patients’ quality of life. Therapeutic responses are not impaired by gender or age [132,133]; however, possible differences between diabetic and non-diabetic patients have not been described. Furthermore, several studies have shown that the clinical application of MSC transplantation in DM type 1 and type 2 has positive metabolic effects, regulating hyperglycemia and insulin resistance [134], which are the key determinants for DCM pathogenesis.

Therefore, evidence points to MSCs as a powerful tool for the treatment of DCM, but larger studies are warranted to determine the ideal cell source, as well as the best route of administration and timing of transplantation, and to confirm the benefits observed in preclinical studies.

## 6. Conclusions

MSCs attenuate cardiac remodeling induced by DM, reducing cardiac fibrosis and improving systolic and diastolic function; thus, MSCs are a promising therapeutic strategy for the prevention of DCM. However, although some clinical trials have shown improvement in cardiac function and quality of life in patients with ischemic and non-ischemic cardiomyopathy, as well as its favorable metabolic effects for patients with DM, it is still necessary to expand the research concerning the effects of MSCs on improving cardiac remodeling and preventing DCM. The lack of an early and accurate DCM diagnosis contributes to the main challenge in the research and development of preventive and effective cell therapy. Despite the promising results found in preclinical studies, animal models are not representative of how the disease appears in diabetic patients, who feature hypertension, coronary artery disease or other cardiovascular disorders. Regarding the therapeutic potential of MSCs, a key point is to identify and optimize the ideal cell source in terms of harvesting, cultivation, expansion, safety and efficacy. DM interferes with MSCs’ properties, amount and function; thus, the preconditioning and genetic manipulation or delivering strategies are recommended to improve their survival and therapeutic efficacy when considering autologous transplantation. Transendocardial delivery of MSCs has been shown to be feasible, safe and beneficial in patients with different types of cardiomyopathies, but further trials with a larger number of patients and a longer follow-up are imperative to assess the long-lasting effects. Furthermore, knowledge of the interference of MSCs and their secreted substances on the signaling pathways is crucial to use their regenerative effect to prevent cardiac remodeling of DCM. Therefore, despite being a highly promising tool for the treatment of DCM, the standardization, quality and consistency of MSC-based therapy still require extensive future research.

## Figures and Tables

**Figure 1 cells-11-00240-f001:**
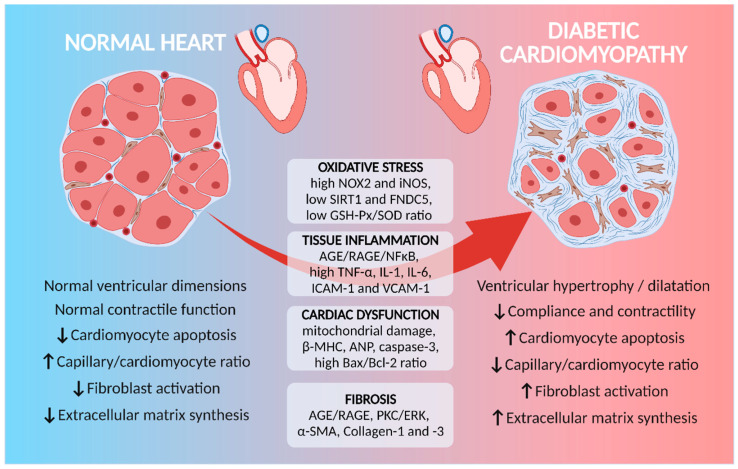
Cardiac functional and morphological alterations in diabetic cardiomyopathy (DCM). Different processes underlie the establishment of DCM, although oxidative stress and tissue inflammation seem to be responsible for maintaining the changes observed in diabetic hearts. Preclinical studies in rodents evidenced alterations in signaling pathways and protein expression related to cardiac dysfunction, which may be relevant targets for the development of new strategies for the treatment of DCM. AGE, advanced glycation end products; ANP, atrial natriuretic peptide; ERK, extracellular signal-regulated kinase; FNDC5, full-length type III fibronectin containing 5; GSH-Px, glutathione peroxidase; ICAM-1, intercellular adhesion molecule-1; IL, interleukin; iNOS, inducible nitric oxide synthase; MHC, myosin heavy chain; NOX2, reduced nicotinamide adenine dinucleotide phosphate-oxidase 2; PKC, protein kinase C; RAGE, receptor for advanced glycation end products; SIRT1, sirtuin 1; SMA, smooth muscle actin; SOD, superoxide dismutase; TNF, tumor necrosis factor; VCAM-1, vascular cell adhesion molecule-1. ↑ increased; ↓ reduced.

**Figure 2 cells-11-00240-f002:**
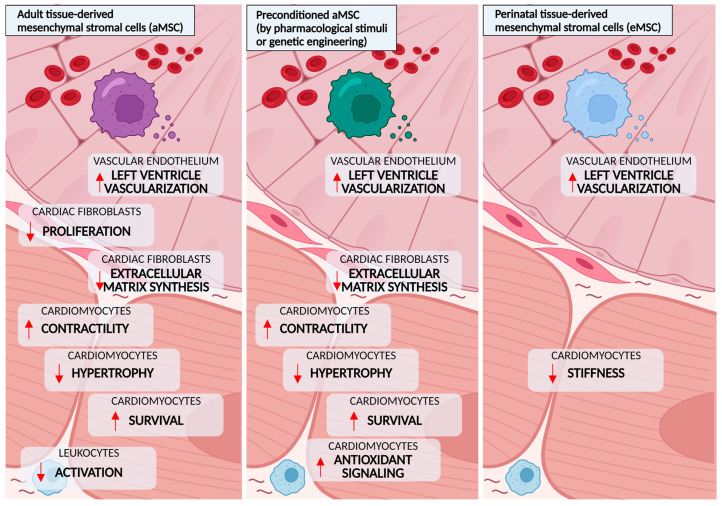
Beneficial effects of mesenchymal stromal cells (MSCs) observed in preclinical models of diabetes. Extracellular vesicle secretion by MSCs derived from adult (bone marrow, adipose tissue) or embryonic tissue-derived (placenta) restores endothelial and cardiomyocyte functions, reverting cardiac hypertrophy and the reduced contractility observed in diabetic animals. In addition to this, by lowering the activation of inflammatory cells and fibroblasts, MSCs reduce myocyte apoptosis and the fibrotic remodeling seen in rodent models.

**Table 1 cells-11-00240-t001:** Summary of findings from preclinical studies.

Agent/Cells	Preconditioning	Model	Effect and/or Mechanism	Ref.
BM-MSCs	-	STZ rats	↓ Cardiac hypertrophy (LV posterior wall thickness and relative wall thickness);	[106]
			↑ Myocardial arteriole density;	
			↑ LV systolic function and FS;	
			↓ LV collagen content;	
			↓ Cardiac expression of MMP-9;	
BM-MSCs	-	HF diet mice	↑/↓ Cardiac contractility (+dP/dt) and relaxation (-dP/dt);	[107]
BM-MSCs	-	STZ+HF/HS diet rats	↓ Cardiac expression of caspase-3;	[108]
			↑ Cardiac expression of 14-3-3, p-Ask1;	
BM-MSCs	-	STZ rats treated with RSV	↓ Cardiac apoptosis (Bax/Bcl2 ratio);	[122]
			↓ Cardiac expression of Wnt3 and β-catenin;	
			↓ Cardiomyocyte hypertrophy;	
			↑ Myocardial capillary density;	
			↑ Cardiac antioxidant defenses (TAC, SOD);	
BM-MSCs	-	STZ rats treated with MET	Attenuated reduction in blood glucose;	[110]
			Attenuated cardiac angiogenesis;	
			Attenuated reduction in LV collagen content;	
AT-MSCs	-	STZ rats	↓ LV wall thinning and dilation;	[113]
			↓ Diastolic dysfunction;	
			↓ Cardiac collagen content and fibrosis;	
			↓ Proliferation of cardiac fibroblasts;	
			↓ Cardiac expression of IL-6, TNF-α, TGF-β;	
			↑ Macrophage polarization to M2 phenotype;	
AT-MSCs	-	STZ+HF diet mice	↓ Blood glucose and cholesterol;	[116]
			↑ LV systolic function (FS and EF);	
			↓ Cardiomyocyte hypertrophy;	
			↓ Cardiac collagen content;	
			↓ Cardiac macrophage number;	
			↓ Cardiac TNF, CXCL15, IL6 mRNA levels;	
			↓ Cardiac expression of IL-1β;	
AT-MSCs (autologous)	-	STZ rats treated with EGCG	↓ Cardiac expression of TGF-β, MMP-9, p-NFκB, COX-2;	[123]
Syndecan-2^+^ BM-MSCs	-	db/db mice	Attenuated cardiac angiogenesis;	[111]
			Attenuated reduction in cardiomyocyte stiffness;	
BM-MSC exosomes	-	STZ rats	↓ LV collagen content;	[109]
			↓ Cardiac TGF-β, Smad2 mRNA levels;	
BM-MSCs	Anoxia	STZ rats	↓ Cardiac hypertrophy (heart weight);	[120]
			↑ LV systolic function (FS);	
			↑ Myocardial capillary density;	
BM-MSCs	RSV	STZ ratstreated with RSV	↓ Cardiac apoptosis (Bax/Bcl2 ratio);	[121,122]
			↓ Cardiac expression of Wnt3, β-catenin and sFRP2;	
			↓ Cardiac collagen content;	
			↓ Cardiomyocyte hypertrophy;	
			↑ Myocardial capillary density;	
			↑ Cardiac antioxidant defenses (TAC, SOD);	
AT-MSCs (autologous)	RSV	STZ rats	↓ Blood glucose;	[115]
			↑ LV systolic function (EF and FS);	
			↑ Cardiac expression of p-IGF1R, p-PI3K, p-Akt, p-AMPK, Sirt1, PGF1α, SOD2;	
			↓ Cardiac expression of ANP, BNP;	
			↓ Cardiac expression of p-Bad, Bcl2, caspase-3;	
			↓ Cardiomyocyte apoptosis (TUNEL);	
BM-MSCs	Adiponectin overexpression	HG-stimulated H9c2 cells	↓ Expression of TGF-β, Smad2/3	[124]
BM-MSCs	Adiponectin overexpression	STZ+HF diet rats	↓ Cardiac hypertrophy;	[124]
			↑ LV systolic function (FS);	
			↓ LV collagen content;	
			↓ Cardiac expression of TGF-β, Smad2/3;	
dm-BM-MSCs	Conditioned medium from HG+H_2_O_2_-stimulated-primary neonatal rat cardiomyocytes	STZ mice	↑ LV systolic function (EF, +dP/dt);	[117]
			↑ LV diastolic function (LVEDP, -dP/dt);	
			↑ Cardiac MEF2c, NKX2.5, GATA-4 mRNA levels;	
			↓ Cardiac NFκB mRNA levels;	
			↓ Cardiac expression of caspase-3;	
			↑ Cardiac VEGF, ANG-1 mRNA levels;	
			↑ Myocardial capillary density;	
			↓ Cardiac collagen content;	
PLX	-	STZ mice	↓ Diastolic dysfunction (-dP/dt, tau);	[126]
			↓ Cardiomyocyte stiffness (p-titin);	
			↑ Cardiac PKA and PKG activities;	
			↑ Cardiac VEGF mRNA levels	
			↑ Myocardial arteriole density;	
			↓ Cardiac IFN-γ and VCAM-1 mRNA levels;	
			↑ Circulating Treg cells;	
PLX	-	HG-stimulated cardiac fibroblasts	↓ Collagen production	[126]
			↓ Myofibroblast transdifferentiation (α-SMA)	

↑, increased; ↓, reduced; ANG-1, angiopoietin-1; AT-MSCs, adipose tissue-derived MSCs; BM-MSCs, bone marrow-derived MSCs; DCM, diabetic cardiomyopathy; dm-BM-MSCs, diabetic mouse-derived BM-MSCs; EF, ejection fraction; EGCG, epigallocatechin-3-gallate; FS, fractional shortening; HF, high fat; HG, high glucose; HS, high sugar; IFN-γ, interferon-γ; IL-6, interleukin-6; IPCs, insulin-producing cells; LV, left ventricle; MEF2c, myocyte-specific enhancer factor; MET, metformin; MMP-9, matrix metalloproteinase-9; MSCs, mesenchymal stromal cells; NFκB, nuclear factor κB; PKA, protein kinase A; PKG, protein kinase G; PLX, placenta-derived MSC-like cells; RSV, resveratrol; RSV-BM-MSCs, BM-MSCs combined with resveratrol; sFRP2, secreted frizzled-related protein; α-SMA, α-smooth muscle actin; SOD, superoxide dismutase; STZ, streptozotocin; TAC, total antioxidant capacity; TGF-β, transforming growth factor-β; TNF-α, tumor necrosis factor-α; VCAM-1, vascular cell adhesion molecule-1; VEGF, vascular endothelial growth factor.

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
