# Peer review of "Mesenchymal Stem Cell Therapy in Diabetic Cardiomyopathy"

_cells, 2022, doi:10.3390/cells11020240_

Round 1

Reviewer 1 Report

The article “Mesenchimal Stem Cells Therapy in Diabetic Cardiomyopathy” by da Silva et al. is a literature review that deals with  mesenchymal stromal therapy as promising approach for the prevention of diabetic cardiomyopathy (DCM). The manuscript offers a comprehensive, balanced, and also critical perspective of the current literature.

Strengths:

- The sections were logically structured/developed.

- Relevant literature is properly discussed in an unbiased manner.

- The Figures are in line with the quality of the current version of this manuscript.

Nevertheless, I suggest to change the head of the first paragraph. Something like “Diabetes- induced molecular mechanisms in cardiac remodeling” would be more appropriate as the paragraph mainly deals with the molecular mechanisms underlying DCM pathogenesis. Herein, the authors should also consider to dedicate a section to the role of epigenetic mechanisms (e.g. role of histone deacetylases) in DCM pathogenesis. To this regard, there is growing evidence showing the contribution of diabetes-induced epigenetic modifications in the development of cardiovascular complications and in  metabolic memory.

A table summarizing the results from the preclinical studies described in the manuscript would be helpful for the reader.

Minor comments:

Page 2, Line 82-90: the sentences are repetitive, please summarize.

Page 8, line 351-354 “BM-MSC isolated from…” the sentence is unclear and needs to be reformulated.

Author Response

1. Nevertheless, I suggest to change the head of the first paragraph. Something like “Diabetes- induced molecular mechanisms in cardiac remodeling” would be more appropriate as the paragraph mainly deals with the molecular mechanisms underlying DCM pathogenesis. Herein, the authors should also consider to dedicate a section to the role of epigenetic mechanisms (e.g. role of histone deacetylases) in DCM pathogenesis. To this regard, there is growing evidence showing the contribution of diabetes-induced epigenetic modifications in the development of cardiovascular complications and in metabolic memory.

As suggested by the reviewer, it has been altered the head of the sections from “2. Diabetes-induced cardiac remodeling” to “2. Molecular mechanisms involved in diabetes-induced cardiac remodeling” and “3. Involvement of inflammation in diabetes-induced cardiac remodeling” to “3. Myocardial inflammation and diabetes-induced cardiac remodeling”. Another section entitled “4. Epigenetic involvement in diabetic cardiomyopathy” was included.

2. A table summarizing the results from the preclinical studies described in the manuscript would be helpful for the reader.

As requested, a table has been added to the manuscript.

Minor comments:

3. Page 2, Line 82-90: the sentences are repetitive, please summarize.

The paragraph was altered as follows (lines 100-105):

“In DM, levels of full-length type III fibronectin containing 5 (FNDC5) has been reported to be reduced which are involved in the cardiomyocytes apoptosis [8, 32]. Overexpression of FNDC5 reduces NOX2 and iNOS activities, improves glucose tolerance, reverses diastolic dysfunction and attenuates cardiac remodeling, demonstrating a significant role of reduced FNDC5 to cardiac dysfunction observed in DM [8].”

4. Page 8, line 351-354 “BM-MSC isolated from…” the sentence is unclear and needs to be reformulated.

The sentence was altered to read as follows (lines 369-374):

“However, some alterations seen in diabetic MSCs can be reverted using myogenic medium preconditioning. Preconditioning of BM-MSCs can improve their proliferative rate and production of cardioprotective and angiogenic factors. This condition can also provide increase of cardiac homing and reduction of cardiac dysfunction and remodeling [117].”

Reviewer 2 Report

COMMENTS TO AUTHORS:

In this review manuscript (cells-1475858), the authors summarized the progress of mesenchymal stem cells in the treatment of Diabetic Cardiomyopathy applications from two aspects of Preclinical and Clinical Approaches. The authors also discussed it is still necessary to promote the effects of MCS on improving cardiac remodeling and preventing DCM. The manuscript is concise and easy to follow. Major comments.

1) The manuscript can be restructured to balance the length of each chapter. The title of the article is "Mesenchymal Stem Cells Therapy in Diabetic Cardiomyopathy", but the authors spent a long text to introduce “Diabetes-induced cardiac remodeling” and “Involvement of inflammation in diabetes-induced cardiac remodeling”. This is not a review of the pathogenesis of Diabetic Cardiomyopathy. The author can first briefly introduce mesenchymal stem cells and diabetic cardiomyopathy and then focus on the rationale of mesenchymal stem cells in the treatment of diabetic cardiomyopathy.

2)  For the part of “Mesenchymal Cells in Diabetes-induced Cardiac Complications” that needs to be highlighted, the author just simply divided it into two aspects, Preclinical and Clinical Approach, which will give the reader a feeling that the description will be too complicated and lengthy. As mentioned in this review, “interesting approaches to increased MSCs therapeutic effects are preconditioning, changing culture conditions or genetically-modifying cells in vitro prior to transplantation.”, It is easier for readers to accept and understand if it is described according to this classification method.

3) In the Conclusion section, the author only briefly mentioned: “it is still necessary to expand the research of the effects of MCS on improving cardiac remodeling and preventing DCM.” This is insufficient. It is recommended to expand this section, e.g., including the discussion on better cell delivery and strategies to enhance therapeutic efficacy, et al. And there are many doubts about the durability and safety of mesenchymal stem cell therapy, which should be addressed before MSCs can be defined as a novel and efficient therapeutic agent in the treatment of Diabetes-induced Cardiac Complications. It might be worthy to focus on past and current clinical trials, summarize the status, the major hurdles/issues, and potential strategies to address the issues.

Author Response

1) The manuscript can be restructured to balance the length of each chapter. The title of the article is "Mesenchymal Stem Cells Therapy in Diabetic Cardiomyopathy", but the authors spent a long text to introduce “Diabetes-induced cardiac remodeling” and “Involvement of inflammation in diabetes-induced cardiac remodeling”. This is not a review of the pathogenesis of Diabetic Cardiomyopathy. The author can first briefly introduce mesenchymal stem cells and diabetic cardiomyopathy and then focus on the rationale of mesenchymal stem cells in the treatment of diabetic cardiomyopathy.

As requested by the reviewer, a brief description of MSCs and MSC therapy was included in the “Introduction” section (lines 43-60).

Mesenchymal stem cells (MSCs) are multipotent cells found in almost all adult tissues with the ability to self-renewal and differentiation, favoring homeostasis and repair, which make them promising tools for regenerative medicine [13]. MSCs show low immunogenicity and home to damaged tissues which can allow allogeneic cell transplantation where can engraft and differentiate promoting repair. However, MSCs have poor engraftment rate and the cellular replacement is complex and limited.  Instead, their therapeutic potential is attributed primarily to the paracrine action of their secretome, which consists of a rich and complex mixture of soluble molecules, such as cytokines, chemokines and growth factors, and extracellular vesicles loaded with proteins, peptides and genetic material (e.g., microRNAs) [14], which can support cell survival and tissue healing. Additionally, the preconditioning with hypoxia, growth factors, cytokines or pharmacological agents and genetic modifications can modulate MSCs survival, proliferation, migration and senescence, in order to preserve them and improve their secretory activity, amplifying their therapeutic efficacy [14]. 

2)  For the part of “Mesenchymal Cells in Diabetes-induced Cardiac Complications” that needs to be highlighted, the author just simply divided it into two aspects, Preclinical and Clinical Approach, which will give the reader a feeling that the description will be too complicated and lengthy. As mentioned in this review, “interesting approaches to increased MSCs therapeutic effects are preconditioning, changing culture conditions or genetically-modifying cells in vitro prior to transplantation.”, It is easier for readers to accept and understand if it is described according to this classification method.

As requested, the section “Preclinical and clinical approach of MSC in diabetes-induced cardiac complications” was subdivided in accordance of the origin and/or pretreatment of MSCs in “Adult tissue-derived MSCs”, “Preconditioning and genetic modification of adult tissue derived-MSCs” and “Perinatal tissues derived-MSCs”.

3) In the Conclusion section, the author only briefly mentioned: “it is still necessary to expand the research of the effects of MCS on improving cardiac remodeling and preventing DCM.” This is insufficient. It is recommended to expand this section, e.g., including the discussion on better cell delivery and strategies to enhance therapeutic efficacy, et al. And there are many doubts about the durability and safety of mesenchymal stem cell therapy, which should be addressed before MSCs can be defined as a novel and efficient therapeutic agent in the treatment of Diabetes-induced Cardiac Complications. It might be worthy to focus on past and current clinical trials, summarize the status, the major hurdles/issues, and potential strategies to address the issues.

The conclusion was expanded in order to include the aspects mentioned by the reviewer. The following text was included (lines 466-484):

“Lack of early and accurate DCM diagnosis brings to the main challenge in research and development of preventive and effective cell therapy. Despite the promising results found in preclinical studies, animal models are not exactly how the disease appears in diabetic patients, who feature hypertension, coronary artery disease or other cardiovascular disorders. Regarding the therapeutic potential of MSCs, a key point is to identify and optimize the ideal cell source in terms of harvesting, cultivation, expansion, safety and efficacy. DM interferes with MSCs properties, amount and function thus, the preconditioning and genetic manipulation or delivering strategies are recommended to improve their survival and therapeutic efficacy when considering autologous transplantation. Transendocardial delivery of MSCs has been shown to be feasible, safe and beneficial in patients with different types of cardiomyopathies, but further trials with larger number of patients and longer follow-up are imperative to assess long-lasting effects. Furthermore, the knowledge of the interference of MSCs and their secreted substances on the signaling pathways is crucial to use their regenerative effect to prevent cardiac remodeling of DCM. Therefore, despite being a highly promising tool for the treatment of DCM, the standardization, quality and consistency of MSC based therapy still require extensive future research.”

Round 2

Reviewer 1 Report

All  comments have been addressed.

Reviewer 2 Report

The author fully complied with the reviewer’s comments and made careful revisions. The quality of the paper has been significantly improved. I am satisfied with the revised article.